# Study on the Quality Variation and Internal Mechanisms of Frozen Oatmeal Cooked Noodles during Freeze–Thaw Cycles

**DOI:** 10.3390/foods13040541

**Published:** 2024-02-09

**Authors:** Xianhui Chang, Hairong Liu, Kun Zhuang, Lei Chen, Qi Zhang, Xi Chen, Wenping Ding

**Affiliations:** 1School of Food Science and Engineering, Wuhan Polytechnic University, Wuhan 430023, China; cxh5286@whpu.edu.cn (X.C.); liuhairong2024@163.com (H.L.); zhuangkunzk@163.com (K.Z.); chenleiy@whpu.edu.cn (L.C.); qizhang21@163.com (Q.Z.); xchen@whpu.edu.cn (X.C.); 2Key Laboratory for Deep Processing of Major Grain and Oil, Ministry of Education, Hubei Key Laboratory for Processing and Transformation of Agricultural Products, Wuhan Polytechnic University, Wuhan 430023, China

**Keywords:** frozen staple food, freeze–thaw cycle, quality changes, microstructure

## Abstract

Frozen staple food, attributed to its favorable taste and convenience, has a promising development potential in the future. Frequent freezing and thawing, however, will affect its quality. This study simulated several freeze–thaw cycles (FTC) that may occur during the cold chain process of frozen oatmeal cooked noodles (FOCN) production to consumption. The quality changes and their mechanisms were elucidated using methods such as differential scanning calorimetry (DSC), low-field nuclear magnetic resonance (LF-NMR), Fourier-transform infrared spectroscopy (FTIR), confocal laser scanning microscopy (CLSM), texture analysis, and sensory evaluation. The freezable water content of the FOCN decreased because of the FTC treatment, and the relative content of total water in FOCN also decreased accordingly. The increase in β-Turn after FTC induced disorder in the secondary structure of proteins, causing the protein microstructure to become loose and discontinuous, which in turn reduced the water-holding capacity of FOCN. Additionally, FTC reduced the chewiness and sensory score of FOCN. This research will contribute a theoretical foundation for optimizing the cold chain process.

## 1. Introduction

Frozen food is a new type of convenience food and is developing towards staple food presently. Noodles, enjoyed by people all over the nation, are one of the most popular and widely consumed traditional Chinese staple foods. Noodles that are currently on the market are primarily separated into four categories: dry noodles, fresh noodles, instant noodles, and frozen noodles [1]. Frozen noodles have the advantages of convenience and good taste. At the same time, the idea of a diet is changing in the minds of modern people from emphasizing satiety to emphasizing healthful eating. Numerous healthy nutrients can be found in oats, including soluble dietary fiber, unsaturated fatty acids, protein, and vitamins [2]. Oats are nutritionally more widely utilized when added to a variety of staple foods. Therefore, to meet the needs of consumers, frozen oatmeal cooked noodles (FOCN), a new product, has been developed. Compared with other traditional noodles, FOCN are attracting increasing consumer attention due to their convenience, nutrition, and health benefits. However, the process of frozen noodles from production to consumers will go through production, storage, transport, distribution, etc., and it is impossible to maintain continuous freezing, which will inevitably lead to temperature fluctuations and the process of freeze–thaw cycles (FTC), which may reduce the quality of frozen noodles and become the key factor to control the quality of frozen noodles.

Many earlier studies concentrated on the impact of processing technology on frozen noodles. Hatcher [3] examined the effects of freezing and reheating on the texture of frozen noodles; Lü et al. [4] studied the cooking method of frozen noodles prior to freezing. Now, we have to think about how the FTC will affect the frozen noodles’ quality. Liu et al. [5] discovered that the cooking loss and broken noodle content of frozen uncooked noodles with poor protein content increased after four FTC of frozen dough. Yang et al. [6] conducted multiple FTC on cooked frozen noodles and found that FTC caused ice crystal damage, resulting in deterioration of the texture of the frozen noodles. These studies show that the number of FTC has a significant impact on the quality of frozen noodles, but the underlying mechanism is still unrevealed, which needs to be further explored.

The demand for cold chain transportation of FOCN is also gradually increasing in the fast-paced contemporary. However, due to the limitations of cold chain technology, it cannot be ensured that noodles remain at −18 °C throughout the four stages from production to storage, storage to transportation, transportation to sales platform, and sales platform to customers [7]. Moreover, numerous problems with food safety and quality can also result from improper temperature regulation during the food distribution process and excessive exposure to uneven temperatures [8]. And several studies have demonstrated that temperature fluctuation in the freezing process leads to the formation of ice crystals in the frozen cooked noodles. Large ice crystals can generate nonuniformity, which adversely affects cryo-preserved foods [9]. Therefore, it is necessary to simulate the effect of FTC on FOCN more realistically. In the production workshop of FOCN, the freezing step is at −30 °C. The storage temperature is generally between −22 °C and −18 °C [10], and −18 °C is selected for cost savings. According to Chinese standard GB/T 24616-2019 [11], the product transportation temperature is taken as −18 °C. After investigating 15 supermarkets, −5 °C is selected as the temperature of the supermarket refrigerator. The temperature of household refrigerators is set to −4 °C to ensure long product storage time and facilitate rapid thawing for consumption. Based on the above analysis, the simulated temperatures for the freezing process, cold storage, cold chain vehicles, supermarket refrigerators, and household refrigerators are −30 °C, −18 °C, −18 °C, −5 °C, and −4 °C, respectively. This study aims to investigate the effects of FTC on the texture, sensory evaluation, freezable water content, water distribution structure, protein secondary structure, and protein network structure of FOCN by simulating the FTC that occurs during these four steps. The internal mechanism of the effects of FTC on FOCN quality is clarified, and a theoretical basis is provided for optimizing the steps that produce FTC.

## 2. Materials and Methods

### 2.1. Materials

FOCN (provided by Nanyang Maixiangyuan Food Co., Ltd., Nanyang, China). All chemicals were of analytical grade. The basic formula mainly includes oat flour (46%), vital wheat gluten (21%), resistant starch (17%) and whole egg powder (13%).

### 2.2. FTC Treatment of Samples

FOCN was taken out of the refrigerator at −30 °C and then thawed at 25 °C for 2 h, which was the 0 FTC. FOCN was frozen at −18 °C for 24 h and then thawed at 25 °C for 2 h, which was the 1 FTC. FOCN was frozen at −18 °C for 24 h and then thawed at 25 °C for 2 h, which was the 2 FTC. FOCN was frozen at −5 °C for 24 h and then thawed at 25 °C for 2 h, which was the 3 FTC. FOCN was frozen at −4 °C for 24 h and then thawed at 25 °C for 2 h, which was the 4 FTC. The scheme of the following works is shown in Figure 1.

### 2.3. Differential Scanning Calorimetry (DSC)

Determination of chilled water content by DSC according to the method reported by Liu et al. [12] with some modification. An amount of 10–15 mg of FOCN was weighed and transferred to a designated DSC crucible. Before conducting the DSC test, the crucible should be properly covered and sealed, and the DSC should be calibrated using an empty crucible as a reference. Before the experiment began, the equilibrium temperature was set to 30 °C, then it was lowered to −30 °C at a rate of 10 °C/min, maintained for 10 min, and finally heated to 30 °C at a rate of 5 °C/min. The water content was measured by using the AACC 44-15A method [13]. An amount of 3–4 g of FOCN with different FTC was weighed, placed in a constant weight aluminum box, then placed at room temperature for 5 min to balance its moisture, and finally wiped off excess moisture on the surface with filter paper and weighed the sample mass. The sample was placed in an oven at 105 °C and baked to constant weight, removed, and weighed its mass. Three parallel determinations were performed on all samples.
(1)$$\mathrm{moisture}\;\mathrm{content}=\frac{m_{2}-m_{1}}{m_{2}}\times 100\%$$
where m_2_ is the mass of the initial noodle (g), and m_1_ is the mass of the noodle after baking to constant weight (g).
(2)$$\mathrm{Freezable}\;\mathrm{water}\;\mathrm{content}=\frac{H_{w}}{H_{i}T_{w}}\times 100\%$$
where *H_w_* is the enthalpy change in water in the sample (J/g), *H_i_* is the enthalpy change in pure water freezing (334 J/g), and *T_w_* is the water content of the sample (%).

### 2.4. Low-Field Nuclear Magnetic Resonance (LF-NMR)

The method of Nawaz et al. [14] was slightly modified. Determination of lateral relaxation time of nuclear magnetic resonance: After cooking for 30 s, FOCN was cut into three 2.5 cm FOCN and placed in parallel in a sample bottle with an inner diameter of 27 mm. During the experiment, FOCN was covered to minimize its water loss. The experiment was conducted in a nuclear magnetic tube with an inner diameter of 40 mm, and the Q-CPMG sequence parameters of the test sequence were set as follows: main frequency SF = 20 MHz, sampling frequency SW = 200 KHz, sampling start point RFD = 0.002 ms, Sampling points TD = 11,996, sampling interval TW = 3000 ms, accumulation number NS = 8, amplification factor PRG = 3, and echo number NECH = 6000. Each group of samples was tested three times. The lateral relaxation data were fitted and inverted 70,000 times using the program provided by the instrument to obtain the T2 relaxation map.

### 2.5. Fourier-Transform Infrared Spectroscopy (FTIR)

Sample preparation: First, the FOCN sample was subjected to 0, 1, 2, 3, and 4 FTC. Then, the freeze–thawed sample was placed in a vacuum freeze dryer for drying, ground, passed through a 120-mesh sieve, and stored in a refrigerator at 4 °C for standby.

The method of Zhang et al. [15] was slightly modified, and FTIR is used to determine the secondary structure of frozen cooked gluten protein. An amount of 1–2 mg of FOCN is ground into a fine powder in agate mortar and mixed evenly with dry potassium bromide powder (approximately 100 mg, particle size 200 mesh). The FOCN sample is placed in a mold and should be compressed into a thin sheet by a tablet press. After that, a comprehensive spectral scan ranging from 400 cm^−1^ to 4000 cm^−1^ is performed. The spectral range from 1600 cm^−1^ to 1700 cm^−1^ is extracted and analyzed using PeakFit_V4.12 software.

### 2.6. Confocal Laser Scanning Microscopy (CLSM)

The method of Silva et al. [16] was slightly modified, the 5 mm FOCN was wrapped in Leica gel and cut into 10 pieces on a frozen microtome. The slices were transferred onto a glass slide and stained with 0.28 g/L fluorescein isothiocyanate (preferentially stained starch) solution for 10 min, followed by staining with 0.013 g/L Rhodamine B (preferentially stained protein) solution for 5 min. The slices were then rinsed with ionized water for 10 s and the excess liquid was absorbed with filter paper. Finally, the stained sections were observed using CLSM with a set of light-emitting diode filters. The excitation/emission wavelength of fluorescein isothiocyanate is 488/518 nm, and the excitation/emission wavelength of Rhodamine B is 568/625 nm. The FOCN slice should be photographed using a 40× objective lens combined with an Olympus FV3000 camera (Olympus Corporation, Tokyo, Japan) and FV31S-SW software (available at https://lifescience.evidentscientific.com.cn/zh/downloads/detail-iframe/?0[downloads][id]=847252002 (accessed on 2 January 2024)).

### 2.7. Texture Properties and Sensory Evaluation

The sensory evaluation team for this experiment consisted of seven members, including three males and four females, who evaluated the appearance, odor, palatability, taste, stickiness, and smoothness of the cooked noodles. Seven trained members from the School of Food Science and Engineering participated in the ranking test. Group members receive training according to ISO (11132:2021) [17] sensory analysis standards [18]. The group consists of seven individuals aged 20 to 30. FOCN at room temperature was re-cooked for 30 s before sensory evaluation. According to the standard procedure, each group member was assigned 10 g of FOCN for sensory evaluation. The panelists were asked to evaluate the noodle within 5 min [19]. All volunteers were informed of the content of the experiment. All volunteers voluntarily participated in this sensory evaluation and agreed to the collection of statistical data. During this study, the rights and privacy of all participants were protected. The experiment did not require approval from an ethics committee because the panel members who tasted the samples did not have any associated risks. The sensory evaluation table is made according to Chinese standard T/CIFST 012-2023 [20] and some references [21,22], as shown in Table 1.

### 2.8. Statistical Analysis

All the experiments were arranged in at least three parallels and the results were expressed as the mean value ± standard deviation via SPSS 21.0 (SPSS Inc., Chicago, IL, USA). The statistically significant differences between the various groups were determined using one way analysis of variance (ANOVA) and Duncan’s multiple range tests (*p* < 0.05). The differences were expressed using lower-case letters (a, b, c, d, e, f, g), with the same letter superscript denoting no significant difference (*p* ≤ 0.05). The data were plotted by using Origin 8.5 (Origin Lab Corp., Northampton, MA, USA).

## 3. Results and Discussion

### 3.1. Effect of FTC on Freezable Water Content of FOCN

Since the amount, distribution, and size of ice crystals are largely dependent on the amount of freezable water, DSC was used to quantitatively quantify and compute the content of freezable water in noodles. A formula can be used to determine the percentage of freezable water by examining the heat flow curve.

As the number of FTC increases, Figure 2 shows that the freezable water content of the FOCN decreases. After 1 FTC, the volume of freezable water decreases by 1.77%. The evaporation of water during the process of ice turning into water is the cause of the notable decrease in the amount of freezable water following 2 FTC [23]. The destructed noodle gel structure as the consequence of repeated formation of ice crystal showed the limited water-holding capability, leading to a decrease in the freezable water content of FOCN. Moreover, the recrystallization of water inside the noodles at the last FTC, which tends to adhere to larger ice crystals, general exhibited a low melting point and a larger free energy [24]. Additionally, the FOCN’s internal architecture is damaged, which leads to mechanical harm that breaks the stable structure of gluten proteins or starch and, as a result, releases some of the water that was previously bound to the starch or gluten proteins [12].

### 3.2. Effect of FTC on Moisture Distribution of FOCN

Noodles have a smooth and viscous taste due to the water distribution during cooking and the cross-linking of starch and protein [25]. The stability of the product’s moisture distribution will be impacted by temperature fluctuations that transpire during the storage and transportation processes, though. Therefore, it is essential to carry out study on how moisture is distributed inside FOCN. The T2 spectra as shown in Figure 3 was inverted from the attenuation curve obtained by LF-NMR technology. Three T2 populations were observed and named starting from the shorter to the longer relaxation time T20 (0.1–1 ms), T21 (1–10 ms), and T22 (10–100 ms), and their relative content were named A20, A21, and A22, respectively. The relatively short T20 was designated strongly bound water, the combination of water molecules with gluten and starch is relatively loose, which is mainly a gluten matrix composed of protein and starch. The T21 component represents weakly bound water. In this state, the combination of water molecules with gluten and starch is relatively loose. The relatively long T22 is free water, mainly distributed in the pores of the gluten network, with the strongest fluidity [26].

T20, T21 and T22 and their relative contents A20, A21 and A22 are shown in Table 2. The mobility of water molecules decreases with decreasing T2 value and increases with increasing value [27]. The FTC treatment would first increase and follow by decreasing the T20 of noodles, but the values of T21 and T22 gradually decreased. This indicated the fluidity of water is weakened. The increase in T20 is due to the melting of water in the FOCN during the FTC, which enhances the mobility of starch molecular chains. The decrease in T20 may be due to the evaporation of a small amount of water in the FOCN during the FTC, resulting in a decrease in free and weakly bound water in the FOCN. The results showed that the value of A20 increased continuously and the value of A22 decreased continuously from 0 FTC to 2 FTC. This demonstrates that throughout the thawing process, the temperature change causes some water to lose, which ultimately results in a decrease in the amount of free water in noodles. After 2 FTC, A20 increases and A21 decreases, indicating that the strongly bound water transformed into free water after the 2 FTC, which was related to the retrogradation of starch during the FTC that weakened the hydrogen bond between water molecules and starch [28]. Additionally, there was no significant change in the value of A21 after the FTC, indicating that the transition of water mainly occurred between tightly bound water and free water. A0 continues to decrease, and A0 decreases significantly from 2 FTC to 3 FTC, which is consistent with the decrease in free water content measured by DSC.

### 3.3. Effect of FTC on the Secondary Structure of Protein in FOCN

Fourier-transform and infrared spectroscopy are combined in the technology known as infrared spectroscopy. This theory holds that a substance is made up of atoms that vibrate at a specific frequency. If the substance being studied is exposed to infrared light, its atoms will absorb the photons and go through a transition, creating an interference spectrogram [29]. A few amide absorption bands make up much of the infrared absorption spectrum of proteins. The secondary structure of proteins can currently be studied using amide I bands [30]. FTIR can be used to quantitatively analyze the composition of different protein secondary structures.

FTIR can be used to obtain the amino groups in protein molecules, where the peaks from 1700 cm^−1^ to 1600 cm^−1^ correspond to the amide I band, representing the stretching vibration of C=O [31]. At present, the identification of amide I band peaks has been widely used, with 1651 cm^−1^ to 1660 cm^−1^ being α-Helix, 1610 cm^−1^ to 1640 cm^−1^ being β- Sheet, 1660 cm^−1^ to 1700 cm^−1^ being β-Turn, and 1640 cm^−1^ to 1650 cm^−1^ being Random [32]. Peakfit software 4.12 (available at https://grafiti.com/peakfit/ (accessed on 3 January 2024)) was used to fit the second derivative of the 1600 cm^−1^ to 1700 cm^−1^ wavelength. The percentage content of each protein secondary structure can be obtained by calculating the peak area of each region, as shown in Table 3. With the increase in the number of FTC, the content of β-Sheet gradually decreases, while the content of β-Turn gradually increases. After the second FTC, the content of β-Sheet was significantly reduced, the content of β-Turn increased significantly, and a report by Choi and Ma [33] showed that compared to β-Turn and random curls, α-Helix and β-Sheet was a highly stable conformation that was relatively ordered, indicating that the stability of the protein secondary structure after FTC will decrease.

### 3.4. Effect of FTC on the Microstructure of FOCN

The viscosity and elasticity of the structure of the protein network are key parts of the highly connected interior of the noodles. The FTC changes the position and size of the ice crystals in the fast-frozen noodles, leading to large voids inside the noodles [34]. CLSM can be used to display the impact of FTC visually and clearly on the protein network structure of FOCN. Figure 4 illustrates how the protein network structure of FOCN changed during the FTC.

From Figure 4, it can be seen that during 0 FTC, the protein network structure of FOCN is tightly arranged and continuous. There is no significant difference in the protein network between 0 FTC and 1 FTC, which is due to the fact that at lower freezing temperatures (−30 °C), the cooling rate is faster and the formation of ice crystals is slower, resulting in the rapid formation of large amounts of ice crystals from the water in FOCN. The microcrystals generated in FOCN are small and consistent, making it difficult to disrupt the internal gluten network structure of FOCN [35]. At 2 FTC, the protein network becomes loose and unevenly distributed in the field of vision. At 4 FTC, the protein network becomes loose and loses continuity, and the degree of disruption of the protein network increases in the field of view. This is because large ice crystals will be formed at a higher freezing temperature such as −4 °C, which makes the inner gluten network of noodles easier to be destroyed [36]. The frozen raw noodles investigated by Liu, Guo and Zhu [5] showed that the microstructure of the frozen raw noodles after the FTC was less continuous than that of the fresh control sample. According to the CLSM results, the structure of the protein network was destroyed during the FTC, the loss of its compactness and continuity, which was primarily related to destruction caused by the formation of large ice crystals within the noodles. These experimental results are consistent with the phenomenon of decreased stability of protein secondary structure in FTIR. The disruption of the protein network structure not only reduces the elasticity, hardness, and chewiness of FOCN, but also increases its cooking loss rate, resulting in a serious decline in consumer experience.

### 3.5. Effect of FTC on Texture Properties and Sensory Evaluation of FOCN

Texture characteristics are considered to be the most important criterion for evaluating the overall quality of pasta [37]. It reduces the influence of human factors and taste variances by objectively communicating the advantages and disadvantages of noodles through data. Texture retention like fresh noodles is thus an important criterion for consumers to accept fast-frozen cooked noodles [35]. Table 4 summarizes the texture parameters and sensory score of FOCN with different FTC.

Table 4 displays the impact of FTC on the sensory score and texture properties of FONC. The hardness, chewiness, and adhesive qualities of FOCN all showed a declining trend and significantly decreased from 2 to 3 FTC. According to several studies, high-hardness noodles construct gluten networks more effectively [25], whereas a decline in hardness is a sign that the gluten network of the noodles was harmed throughout the freezing and thawing process. The sensory score significantly dropped during the 2 FTC, which suggests that the quality of FOCN has declined. The relationship between texture properties and sensory evaluation was examined using the Pearson correlation coefficient [38]. The correlation coefficients of 0.9839, 0.9704, and 0.9759 between the sensory score and the hardness, chewiness, and gumminess of FOCN, respectively, indicating a strong correlation between them. This experimental result is consistent with the experimental result of Sozer et al. [39].

Sensory evaluation is an important evaluation method for noodle quality, and consumer acceptance can directly reflect the quality of noodles [40]. The results of sensory evaluation are shown in Figure 5. The total scores for appearance, odor, palatability, taste, stickiness, and smoothness are 20, 10, 20, 20, 15, and 15, respectively. There was no significant difference in the appearance and odor changes in FOCN after FTC, indicating that FTC have a small impact on the appearance and odor of FOCN. From Table 4 and Figure 5, the sensory score of FOCN with 0 FTC is the highest, with the best smoothness and stickiness evaluation, and there is a significant difference from samples with 2, 3, and 4 FTC. This is due to the retrogradation of starch after FTC, resulting in increased stickiness on the surface of noodles due to the leakage of starch granules, leading to a decrease in smoothness and stickiness scores [34], and a corresponding decrease in sensory scores.

## 4. Conclusions

Our research simulated the impact of FTC on FOCN quality during the cold chain process. The results showed that the FTC had little effect on FOCN quality during production and storage, but from the beginning of the transportation process, the deterioration caused by FTC increased, resulting in a decrease in the content of freezable water in FOCN. The protein network structure became discontinuous, and the texture and sensory properties also decreased accordingly. This study will provide a theoretical basis for cold chain transportation of food. Through analysis of the experimental results, we recommend eliminating FTC within controllable areas such as the production process and the storage process, while optimize the FTC that occur in other processes.

## Figures and Tables

**Figure 1 foods-13-00541-f001:**
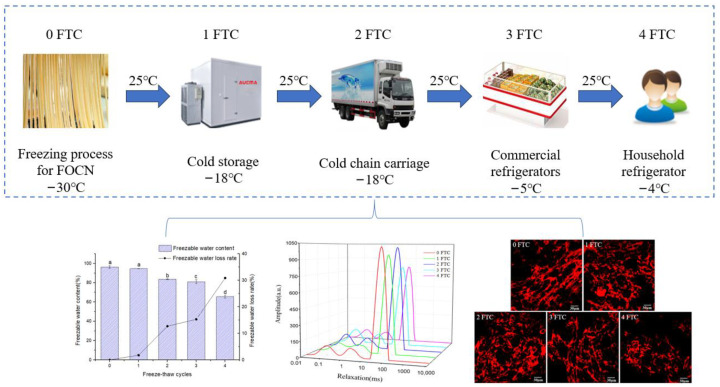
Scheme of experimental design. The effect of 4 FTC on FOCN was simulated, and the samples were tested by differential scanning calorimetry (DSC), low-field nuclear magnetic resonance (LF-NMR) and confocal laser scanning microscopy (CLSM). Vertical bar indicates standard deviation of the mean (n = 3). Different letters represent significant difference (*p* < 0.05).

**Figure 2 foods-13-00541-f002:**
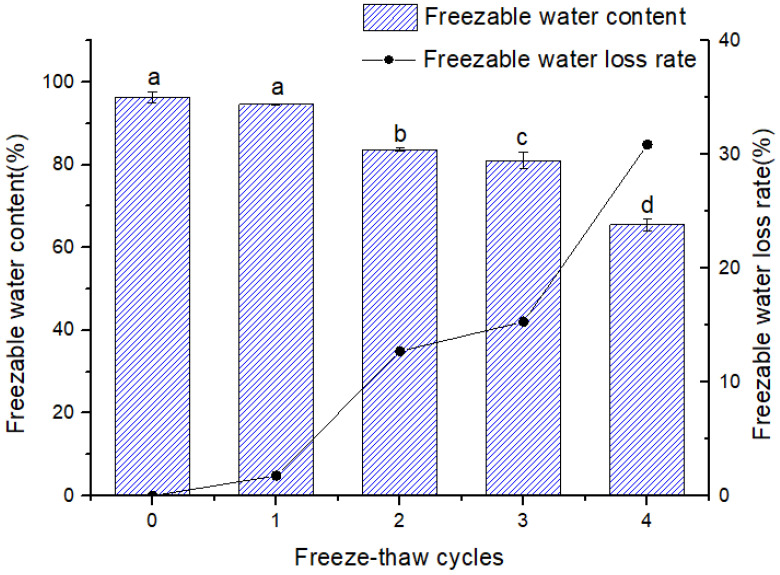
The effect of freezing water content and freezing water loss rate under different FTC. Each experiment was carried out in triplicate. Values with different lower differ significantly among samples (*p* < 0.05).

**Figure 3 foods-13-00541-f003:**
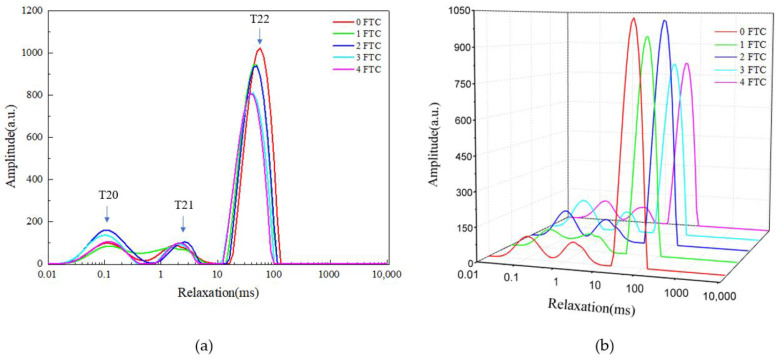
The LF-NMR spin relaxation (T_2_) test of FOCN during FTC. (**a**) Two-dimensional diagram of T2 relaxation distribution curve. (**b**) Three-dimensional diagram of T2 relaxation distribution curve. 0 FTC, 1 FTC, 2 FTC, 3 FTC, and 4 FTC represent 0, 1, 2, 3, and 4 freeze–thaw cycles, respectively. T20: relaxation time of strongly bound water. T21: relaxation time of weakly bound water. T22: relaxation time of free water.

**Figure 4 foods-13-00541-f004:**
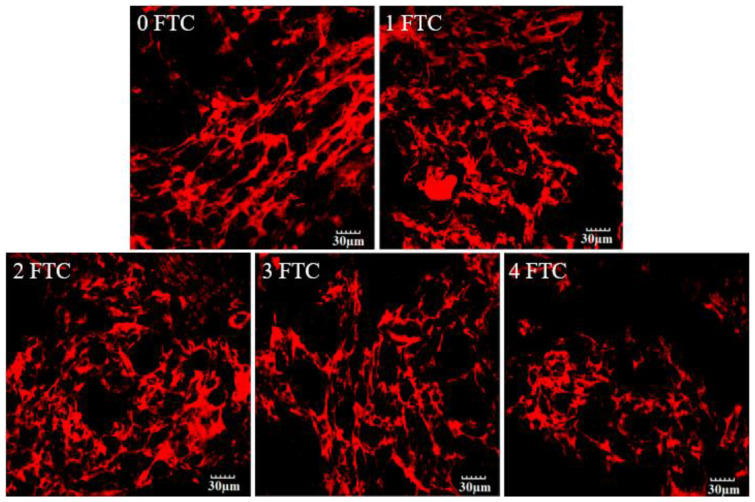
Effect of FTC on rhodamine-labelled gluten microstructure observed in the fluorescence mode. 0 FTC, 1 FTC, 2 FTC, 3 FTC, and 4 FTC represent 0, 1, 2, 3, and 4 freeze–thaw cycles, respectively.

**Figure 5 foods-13-00541-f005:**
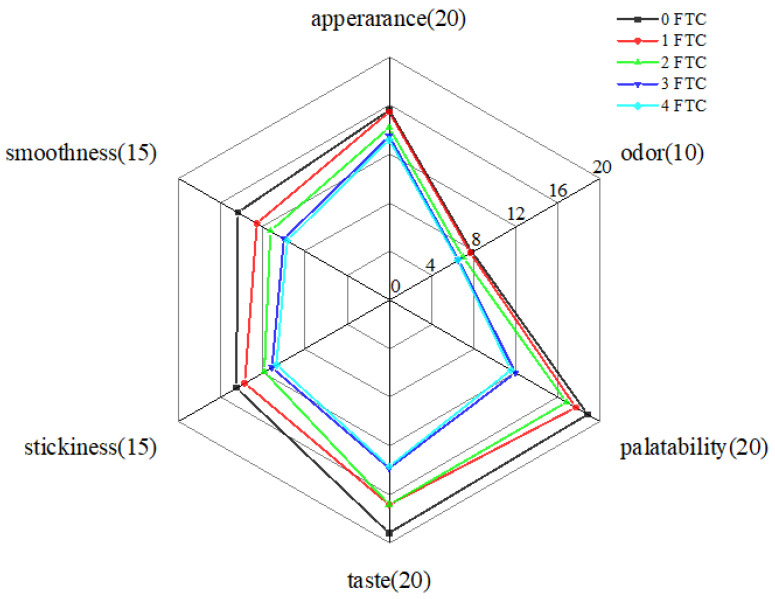
Sensory analysis of FOCN with different FTC. 0 FTC, 1 FTC, 2 FTC, 3 FTC, and 4 FTC represent 0, 1, 2, 3, and 4 freeze–thaw cycles, respectively.

**Table 1 foods-13-00541-t001:** Sensory evaluation of frozen oatmeal cooked noodles.

Sensory Attributes	Score	Definition
Appearance	20	The noodles have a bright and smooth surface, scoring 16–20 points.
		The noodles have a generally bright and slightly rough surface, scoring 11–15 points.
		The noodles have a dull and rough surface, scoring 1–10 points.
Odor	10	The noodles have a strong oat flavor and no off-odors, scoring 8–10 points.
		The noodles have a strong oat flavor and less off flavor, scoring 5–7 points.
		The noodles have no aroma of oats and have a strange smell, scoring 1–4 points.
Palatability	20	The noodle is moderately soft and hard, scoring 16–20 points.
		The noodles are slightly soft or slightly hard, scoring 11–15 points.
		The noodles are too soft or too hard, scoring 1–10 points.
Taste	20	The noodles taste oat-like and have no off-odors, scoring 16–20 points.
		The noodles taste without an oat flavor and have no off odors, scoring 11–15 points.
		the noodles taste sour or have an off flavor, scoring 1–10 points.
Stickiness	15	The noodles are refreshing and not sticky, scoring 11–15 points.
		The noodles are slightly refreshing and slightly sticky, scoring 6–10 points.
		The noodles are not refreshing and sticky, scoring 1–5 points.
Smoothness	15	The noodles are smooth when they enter the mouth, scoring 11–15 points.
		The noodles are slightly smooth when they enter the mouth, scoring 6–10 points.
		The noodles are not smooth when they enter the mouth, scoring 1–5 points.

**Table 2 foods-13-00541-t002:** Effects of FTC on moisture distribution of FOCN (*n* = 3).

FTC	T20 (ms)	T21 (ms)	T22 (ms)	A20 (%)	A21 (%)	A22 (%)	A0 (%)
0	0.32 ± 0.09 ^b^	8.09 ± 1.73 ^a^	74.97 ± 37.49 ^a^	12.60 ± 1.64 ^ab^	7.96 ± 0.76 ^a^	79.44 ± 0.93 ^a^	11096.16 ± 96.55 ^a^
1	0.34 ± 0.09 ^b^	4.67 ± 1.67 ^b^	69.53 ± 34.90 ^ab^	14.57 ± 3.56 ^ab^	8.61 ± 1.05 ^a^	76.82 ± 2.55 ^ab^	10887.44 ± 425.25 ^a^
2	0.54 ± 0.12 ^a^	3.52 ± 0.85 ^b^	65.21 ± 32.60 ^b^	17.22 ± 1.11 ^a^	7.57 ± 0.55 ^a^	75.21 ± 0.93 ^b^	10723.10 ± 145.59 ^a^
3	0.25 ± 0.02 ^b^	5.34 ± 1.80 ^b^	66.70 ± 34.09 ^b^	13.28 ± 2.80 ^ab^	9.12 ± 1.34 ^a^	77.60 ± 1.70 ^ab^	9765.10 ± 227.48 ^b^
4	0.29 ± 0.08 ^b^	3.33 ± 0.86 ^b^	62.35 ± 31.18 ^b^	10.63 ± 3.46 ^b^	9.60 ± 2.19 ^a^	79.78 ± 1.32 ^a^	8916.46 ± 158.78 ^c^

A20, A21 and A22 represent peak areas of T20, T21 and T22, respectively. Values are expressed as the means ± standard deviation. The different letters in the same column indicate significant differences (*p* < 0.05).

**Table 3 foods-13-00541-t003:** Effect of FTC on the secondary structure of protein in FONC (*n* = 3).

FTC	Secondary Structure (%)	
	β-Sheet	β-Turn
0	54.11 ± 0.02 ^a^	45.88 ± 0.02 ^b^
1	53.55 ± 0.00 ^a^	46.21 ± 0.00 ^b^
2	51.79 ± 0.01 ^ab^	48.19 ± 0.01 ^ab^
3	50.42 ± 0.02 ^b^	49.52 ± 0.03 ^a^
4	49.52 ± 0.00 ^b^	50.37 ± 0.01 ^a^

Values are expressed as the means ± standard deviation. The different letters in the same column indicate significant differences (*p* < 0.05).

**Table 4 foods-13-00541-t004:** Effect of FTC on texture properties and sensory score of FOCN (*n* = 3).

FTC	Hardness/g	Adhesiveness/(g∗s)	Chewiness/g	Gumminess/g	Sensory Score
0	1508.44 ± 104.55 ^a^	72.34 ± 22.26 ^a^	840.49 ± 51.98 ^a^	1078.04 ± 83.59 ^a^	90.33 ± 1.53 ^a^
1	1484.14 ± 66.66 ^a^	100.75 ± 36.96 ^a^	791.35 ± 79.64 ^a^	1063.45 ± 52.45 ^a^	84.00 ± 1.00 ^b^
2	1422.13 ± 32.25 ^a^	90.21 ± 8.80 ^a^	789.30 ± 80.75 ^a^	1030.17 ± 70.51 ^a^	78.00 ± 1.00 ^c^
3	1263.75 ± 99.32 ^b^	90.64 ± 23.18 ^a^	697.55 ± 52.37 ^b^	913.67 ± 82.99 ^b^	67.00 ± 2.00 ^d^
4	1244.82 ± 22.89 ^b^	94.06 ± 17.00 ^a^	658.55 ± 49.94 ^b^	901.11 ± 28.03 ^b^	65.33 ± 2.08 ^d^

Values are expressed as the means ± standard deviation. The different letters in the same column indicate significant differences (*p* < 0.05).

## Data Availability

The original contributions presented in the study are included in the article, further inquiries can be directed to the corresponding author.

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
