# Peer review of "Study on the Quality Variation and Internal Mechanisms of Frozen Oatmeal Cooked Noodles during Freeze–Thaw Cycles"

_foods, 2024, doi:10.3390/foods13040541_

Round 1

Reviewer 1 Report

Comments and Suggestions for Authors

Comments on the Quality of English Language

Minor editing of English language required.

Author Response

On behalf of my co-authors, we thank you very much for reviewing our manuscript, we appreciated the editor and reviewers for their constructive comments and suggestions of the manuscript entitled: “Study on the quality variation and mechanism of frozen oatmeal cooked noodles during freeze-thaw cycle” (foods-2832625).

We carefully studied the reviewers' comments and made modifications using "marked blue" in the manuscript (word file). Each reviewer's item-by-item response was specifically processed and uploaded to the submission system.

Thank you very much for your re-consideration.

Sincerely,

Xianhui Chang

Reviewer 2 Report

Comments and Suggestions for Authors

Comments on the Quality of English Language

Author Response

(The authors gave the same response as above.)

Round 2

Reviewer 1 Report

Comments and Suggestions for Authors

The manuscript has been substantially improved. I suggest to the authors a careful check of the English.

Comments on the Quality of English Language

I suggest to the authors a careful check of the English.

Author Response

On behalf of my co-authors, we thank you very much for reviewing our manuscript, we appreciated the editor and reviewers for their constructive comments and suggestions of the manuscript entitled: “Study on the quality variation and mechanism of frozen oatmeal cooked noodles during freeze-thaw cycle” (foods-2832625).

We carefully studied the reviewers' comments and made modifications using "marked red" in the manuscript (word file). Each reviewer's item-by-item response was specifically processed and uploaded to the submission system.

Thank you very much for your re-consideration.

Sincerely,

Xianhui Chang
